# Genome-Wide Characterization of *AspATs* in *Populus*: Gene Expression Variation and Enzyme Activities in Response to Nitrogen Perturbations

**Tao Su** [1,2], **Mei Han** [1,*], **Jie Min** [1], **Dan Cao** [1], **Guangqing Zhai** [3], **Huaiye Zhou** [1,2], **Nanyue Li** [3] **and Mingzhi Li** [4]

[1] Co-Innovation Center for Sustainable Forestry in Southern China, College of Biology and the Environment, Nanjing Forestry University, Nanjing 210037, China; tao.su@cos.uni-heidelberg.de (T.S.); mj121451423@outlook.com (J.M.); qiuqiu121156291@outlook.com (D.C.); z15325487960@163.com (H.Z.)

[2] Key Laboratory of State Forestry Administration on Subtropical Forest Biodiversity Conservation, Nanjing Forestry University, Nanjing 210037, China

[3] College of Forest, Nanjing Forestry University, Nanjing 210037, China; ZGQ1324473961@outlook.com (G.Z.); LNY819343991@outlook.com (N.L.)

[4] Genepioneer Biotechnologies Co. Ltd., Nanjing 210014, China; Limz@genepioneer.com

[*] Correspondence: sthanmei@njfu.edu.cn; Tel.: +86-1589-598-9551

**Abstract:** Aspartate aminotransferase (AspAT) catalyzes a reversible transamination reaction between glutamate and oxaloacetate to yield aspartate and 2-oxoglutarate, exerting a primary role in amino acid biosynthesis and homeostasis of nitrogen (N) and carbon metabolism within all cellular organisms. While progress in biochemical characterization of AspAT has been made for decades, the molecular and physiological characteristics of different members of the *AspAT* gene family remain poorly known particularly in forest trees. Here, extensive genome-wide survey of AspAT encoding genes was implemented in black cottonwood (*Populus trichocarpa* Torr. & A. Gray), a model species of woody plants. Thorough inspection of the phylogenies, gene structures, chromosomal distribution, *cis*-elements, conserved motifs, and subcellular targeting resulted in the identification of 10 *AspAT* isogenes (*PtAspAT1-10*) in the *Populus* genome. RNA-seq along with quantitative real-time polymerase chain reaction (qRT-PCR) validation revealed that *PtAspATs* displayed diverse patterns of tissue-specific expression. Spatiotemporal expressions of homologous *AspATs* in the poplar hybrid clone 'Nanlin895' were further evaluated, showing that gene expressions varied depending on source-sink dynamics. The impact on *AspAT* transcripts upon N starvation and seasonal senescence showed the upregulation of five *AspAT* in leaves concurrent with drastic downregulation of six or more *AspATs* in roots. Additionally, marked reductions of many more *AspATs* transcripts were observed in roots upon N excess. Accordingly, AspAT activities were significantly suppressed upon N starvation by an in-gel assay, prompting the argument that enzyme activity was a more direct indicator of the growth morphology under a N stress regime. Taken together, the expression profiling and enzyme activities upon stress cues provide a theoretical basis for unraveling the physiological significance of specific gene(s) in regulation of N acquisition and remobilization in woody plants.

**Keywords:** aspartate aminotransferase (AspAT); nitrogen acquisition and metabolism; nutrients feeding; senescence; post-transcriptional regulation; poplar

## 1. Introduction

Nitrogen (N) is an essential nutritional element for plants and their interaction with the environment. Aspartate aminotransferase (AspAT; EC 2.6.1.1), a pyridoxal-5′-phosphate-dependent

enzyme, catalyzes reversible transamination reactions between aspartate (Asp) and 2-oxoglutarate (2-OG), leading to formation of glutamate (Glu) and oxaloacetate (OAA) [1]. AspAT exerts a primary role in the regulation of metabolic balance between N and carbon (C). AspAT-mediated biosynthesis of essential amino acids and their corresponding metabolites are incorporated into numerous metabolic and physiological processes that facilitate plant growth and development as well as defense response [2].

Asp serves as a substrate for asparagine (Asn) synthesis in cytosols and is a precursor involved in the biosynthesis of four key essential amino acids; methionine, lysine, threonine, and isoleucine in plastids [2,3]. The former two have been well explored to function as the most limiting amino acids in grains of cereals and legume crops, which may be utilized as major nutrient resources for human and livestock feed [4,5]. Moreover, Asp has been reported to play primary roles as N donors and carriers, supplying amide precursors for the synthesis of N transport compounds, particularly when the recycling of C skeletons are confined in roots under dark conditions, suggesting its association with C shuttling within the intercellular compartments [6,7]. As one of the AspAT catalyzed metabolites, OAA is a pivotal intermediate of the tricarboxylic acid cycle (TCA), gluconeogenesis, glyoxylate cycle, and urea cycle [8]. 2-OG was postulated to be an essential molecule for the biosynthesis of ATP and reducing equivalent (e.g., $NAD^+$/NADH), and a component for the G protein signaling system [9]. 2-OG has also been considered as an obligatory cofactor of many dioxygenases involved in DNA demethylation, epigenetic response, longevity, and maintenance of stem cells [10].

Evolutionary analyses of genomic and protein sequences in various cellular organisms have suggested that AspAT is a small gene family that is widely distributed in microbes, animals, and plants [2]. Based on sequence identities from various organisms, AspAT has been categorized into two families, Iα and Iβ [3]. While Iα contains AspAT from eubacteria and eukaryotes; Iβ includes prokaryotic AspAT/PAT, a bifunctional enzyme with both classic AspAT activity and prephenate amionotransferase (PAT) activity, which shares 15% identities of amino acid sequence with Iα [3]. In silico prediction of the protein model implicated that plastidic AspATs are homodimers, consisting of two identical monomers with a molecular size of approximately 45kDa [6]. Both prokaryotic- and eukaryotic-type AspAT holding a putative signal peptide at N terminus have been utilized for targeting plastids [11]. Early lines of biochemical studies by analyzing the enzyme activities in native gels indicated that AspATs might be localized to various subcellular compartments including cytosols, mitochondrion, plastids, and peroxisomes [6,12]. However, experimental analyses of the subcellular targeting of different *AspATs* were lacking, prompting an urgent need to track their compartmentations in vivo by visualizing the signals from the expressed fluorescence fusion protein.

Beyond the subcellular targeting within subfamilies, emerging research has revealed significant expression variations of *AspAT* isoforms among various tissues as well as in response to environmental stimuli. In *Arabidopsis*, RNA gel blot analysis showed that cytosolic *AtASP2* was expressed predominantly in the roots, whereas transcripts of another cytosolic isoform *AtASP4* were not detected, whereas the mitochondrial *AtASP1* showed extensive expression differences in all *Arabidopsis* tissues [1]. *AtASP3* (*YLS4*) mRNA was remarkably rich in senescent rosette leaves [13]. The bifunctional *AtAspAT/PAT* has been reported to be specifically expressed in green tissues according to immunoblotting [14]. A maize plastidic *ZmAspAT* showed expression abundance in leaves, particularly in the youngest fully expanded leaf blade [15–17]. Expressions of cytosolic *AspATs* in both *Panicum miliaceum* (C4) and rice (C3) have been identified in an organ-specific and light-dependent manner; however, expressions of the mitochondrial *AspATs* in C4 were completely varied from its C3 counterparts [18]. Interestingly, it was found that N nutrient status influenced transcripts accumulation of *AspATs* and enzyme activities of different cellular components of *AspAT* in crops [16,19]. The mitochondrial *ZmAspAT1.3* and plastidic *ZmAspAT/PAT* are markedly promoted by N deprivation and excessive feeding [15]. Additionally, phytohormones may crosstalk with N signaling that can also significantly impact on transcripts of *AspATs*. A previous report showed that the aspartate transaminase (*AAT2*)-like *AspAT* was remarkably induced by exogenous treatments of jasmonic acid in the poplar *P. nigra* [20]. Exogenous feeding of N metabolite GABA (γ-aminobutyric acid) under

low C conditions had a strong inductive effect on the cytosolic *AtASP2* in *Arabidopsis* seedlings [21]. A global transcriptomic overview upon the alternated N levels in the poplar *P. simonii* revealed that N starvation led to suppression of *ASP2* and *ASP5* transcripts that may be correlated with the marked reduction of Arg concentration in roots [22]. Additionally, drought stress downregulated *PvAAT-2* in root nodules of a common bean [23]. Furthermore, transcriptomic analysis showed up-and-down regulations of *AspAT* isoforms in response to drought stress in poplar hybrids (*P. deltoides* × *P. nigra*) [24]. Altogether, variation of spatiotemporal expression and dynamic responses to environmental cues signify the functional divergence and the physiological importance of *AspATs* in the regulation of C and N metabolism, which facilitates plant growth, development, and stress acclimation.

Nevertheless, the functional roles of AspAT in vivo have hitherto been extensively explored in *Arabidopsis* and tobacco. In total, in *Arabidopsis*, five eukaryotic genes (*AspAT1-AspAT5*) encoding AspAT and one prokaryotic type *AspAT/PAT* have been characterized [14,25]. Loss-of-function mutants of cytosolic *AspAT2* (*AAT2*) showed the phenotype of retarded growth concurrent with drastic decreases of Asp and Asn contents in the phloem of dark-grown plants, indicating that the cytosolic *AspAT2* determines functions in the synthesis of Asp/Asn in a light-dependent manner [7,26]. Overexpression of cytosolic or plastidic *OsAspAT* in rice resulted in elevated enzyme activities as well as levels of free amino acid in grains, further supporting its role in N metabolite regulation [5]. A recent report demonstrated that overexpression of *AtAspAT2* in *Arabidopsis* led to the formation of more spreading lesions after infection with pathogenic fungi, suggesting its involvement in the plant defense response, possibly through modification of production of defense compounds [27]. It is worth remarking that the bifunctional isozyme AtAspAT/PAT is co-localized with AtAspAT5 in plastids [14,26,28]; however, it has been shown that these two isozymes exhibited distinct functions in plants [2,14,26]. Ectopic expression of soybean plastidic *GmAspAT5* in *Arabidopsis* improved accumulation of Gln and Asn in seeds, whereas silencing a bifunctional prokaryotic *NbPAT* in tobacco caused significant suppressions of plan growth and remarkable decreases in chlorophyll and lignin contents, along with extremely low levels of Asp, highlighting essential roles of both types of *AspAT* in N metabolism [3,29].

*Populus* is a model organism for molecular biology studies in perennial woody plants and forestry, being of high value and superiority for plantation, biomass production, and potential ecological sustainability [30,31]. In spite of the increasing advances made in AspAT biochemistry properties and enzyme kinetics in several plant species, to date, comprehensive analyses of AspAT encoding genes at molecular genetic and physiological levels are rarely documented in woody plants including *Populus*. Unravelling the physiological significance of a specific gene family has been hampered by the lack of sufficient molecular basis and background. Here, to gain basic knowledge on the molecular and physiological aspect of *AspATs* in poplar, we initially performed a genome-wide survey of AspAT genes within the latest genome annotation in *Populus trichocarpa* through a bioinformatics approach. Expression of *PtAspATs* isoforms in tissues upon feeding of various N sources were analyzed in *P. trichocarpa* by both transcriptomic sequencing (RNA-seq) and qRT-PCR validation. Furthermore, 10 *AspATs* candidates were identified in the poplar hybrid 'Nanlin895' according to homologs in *P. trichocarpa*. We then profiled the spatiotemporal expression of *AspATs* in specific tissues and in response to perturbations of N nutrients and seasonal senescence. Additionally, under the same conditions, enzyme activities from different components (cytosolic, plastidic, and mitochondrial AspAT isozyme) were determined through a native gel assay. Molecular and physiological analysis of *AspAT* family genes revealed their potential functions within different metabolic processes and pathways of amino acid biosynthesis, providing important clues to target specific genes via genetic engineering technologies.

## 2. Materials and Methods

### 2.1. Plant Cultivation and N Treatments

*P. trichocarpa* (genotype 'Nisqually-1') was cultured on standard wood plant medium (WPM) medium containing 20 g L$^{-1}$ sucrose and 8 g L$^{-1}$ plant agar (Biofroxx) in growth chambers, with a

temperature cycling between 22 °C (night) and 26 °C (day) under long-day conditions (16 h light/8 h dark, 20 μmol m$^{-2}$ s$^{-1}$) according to a previous report [32]. Unless otherwise specified, plants cultured for six weeks were harvested for qRT-PCR analysis. Poplar clones '*Nanlin895*' are hybrids of *P. deltoids* × *P. euramericana* developed at Nanjing Forestry University. Plants were cultured in vitro (25 °C, 16/8 h day/night photoperiod, 20 μmol m$^{-2}$ s$^{-1}$) on MS medium with 20 g L$^{-1}$ sucrose, 0.1 mg L$^{-1}$ IBA, and solidified with 8 g L$^{-1}$ plant agar. After a culturing for 4 weeks, seedlings were transferred to standard pots with a mixture of vermiculite: perlite: peat (1:1:3, v/v/v). Afterwards, the plants were transferred to pots with mixtures of sand: perlite (2:1, v/v) for N treatments according to a previous study [22]. After a growth period of 2–4 months, respective tissues were harvested for qRT-PCR and enzyme assay. Seasonal senescence leaves were harvested from plants grown in a green house. Mature leaves were sampled from three time points based on senescent phenotypes observed according to a previous study [33] and frozen samples were ground in liquid nitrogen and subjected to RNA and protein extraction, followed by qRT-PCR analyses and determination of enzyme activity.

## 2.2. Sequence Available and Identification of AspATs

To search for AspAT homologs in poplar, previously identified *AspATs* in *Arabidopsis* and rice were collected from the Arabidopsis Information Resource (TAIR) database (http://www.arabidopsis.org/) and Rice Genome Annotation Project (RGAP) (http://rice.plantbiology.msu.edu/index.shtml). Using known AspAT protein sequences as queries to seed a BLAST against *P. trichocarpa* genome assembly (v3.0) available at the JGI gene catalog (Phytozome v12.1, https://phytozome.jgi.doe.gov/pz/portal.html) with the *E*-value cutoff set as 1e-5. The respective genomic and protein sequence of *PtAspAT* candidates were verified with the Pfam (http://pfam.xfam.org/) by HMMER program (3.1b2) and the National Center for Biotechnology Information (NCBI) (https://www.ncbi.nlm.nih.gov/) to ascertain the presence of Hidden Markov Model (HMM) profiles (PF00155.20) that correspond to Aminotran_1_2. Incomplete protein sequences with short length (<300aa) were eliminated. The remaining *PtAspATs* isogenes were further confirmed by manual removal of the redundant sequences that showed more than 97% identities of amino acid sequence among the different databases. Protein sequences of AspATs in *Eucalyptus grandis* (Eucgr), *Glycine max* (Glyma), *Grossypium raimaondii* (Goari), *Medicago truncatula* (Medtr), *Sorghum bicolor* (Sobic), *Solanum lycopersicum* (Solyc), and *Zea mays* (Zm) were retrieved from Phytozome v12.1. Protein sequences of AspATs in *Nicotiana benthamiana* (Nb) and *Pinus taeda* (Pta) were obtained from the literature [2,12].

## 2.3. Phylogenetic Analysis and Calculation of Ka/Ks

Multiple sequence alignment was conducted by the Clustal X2.1 software with default settings. A phylogenetic tree was constructed in the Molecular Evolutionary Genetics Analysis (MEGA7) (https://www.megasoftware.net/), using neighbor-joining method with 1000 bootstrap replicates [34]. Duplication events were analyzed based on the synteny blocks from the Plant Genome Duplication Database (PGDD) (http://chibba.agtec.uga.edu/duplication/). Reciprocal BLAST was also carried out to establish the genetic relationship between the gene pairs. Hits with E-value ≥ 1e-05 and at least 80% homology were considered significant. For estimating the synonymous (Ks) and non-synonymous (Ka) substitution rates, the corresponding amino-acid and cDNA sequences of paralogous and orthologous PtAspAT proteins were analyzed using the Ka/Ks calculator [35]. Divergence time (T) was calculated using the formula T = Ks/2λ × 10$^{-6}$ (λ = 9.1 × 10$^{-9}$ for *P. trichocarpa*) million years ago (Mya), according to a previous report [36].

## 2.4. Genomic Distribution, Gene Structure, Protein Motif, and Cis-Regulatory Element

The chromosomal distribution of *Populus AspAT* genes was obtained from the the Populus Genome Integrative Explorer (PopGenIE) database (http://popgenie.org/chromosome-diagram), and the location images were drawn with MapInspect software (http://www.softsea.com/review/MapInspect.html). The genomic structure was deduced by comparing the coding sequence (CDS) and corresponding

genomic sequence using the Gene Structure Display Server (GSDS) [37]. The conserved motifs were identified by the Multiple Em for Motif Elicitation (MEME) program (http://meme-suite.org/index.html) with default settings except that the maximum numbers and widths of motifs were set to 6 and 50, respectively [38]. Annotations of the identified functional motifs were depicted by NCBI Conserved Domains Database (CDD) (https://www.ncbi.nlm.nih.gov/cdd) and ScanProsite (http://prosite.expasy.org/scanprosite/) [39]. Upstream 1500 bp regions were used to search for the cis-acting regulatory elements in the Plant Cis-acting Regulatory Elements (PlantCARE) database (http://bioinformatics.psb.ugent.be/webtools/plantcare/html/). Putative TF binding sites within the promoter regions were analyzed in the Plant Transcription Factor Database (PlantTFDB 4.0) (http://planttfdb.cbi.pku.edu.cn/).

### 2.5. Transcriptomic Sequencing and Expression Analysis

For spatial expression analysis in *P. trichocarpa*, transcriptomic sequencing (RNA-seq) data were collected from Phytozome (v12.1) for 18 various tissues including five buds (pre-dormant I, pre-dormant II, early dormant, late dormant, and fully open buds), three leaves (immature, young, and first fully expanded leaves), two stems (node and internode), two roots (root tip and standard root), three developmental stages of male flowers (Early I, Early II, and Mid), and three developmental stages of female flowers (Early, Late, and Receptive). Gene transcript levels in various tissues were valued by fragments per kilobase (kb) of exon model per million mapped reads (FPKM) and presented in heat map. Additionally, RNA-seq data for the phloem, differentiating stem xylem, and three cell types (fiber, vessel, and ray cells) of *P. trichocarpa* were retrieved from Gene Expression Omnibus (GEO) at NCBI (GSE81077) according to a previous report [40]. The heat map was generated by using the pretty heatmaps (Pheatmap) package in R (v3.4.0) with the log2 value of FPKM (Figure S1).

For the qRT-PCR assay, RNA extraction and cDNA synthesis were performed according to a previous report [41]. Total RNA was extracted using the RNeasy Plant Mini Kit (Tiangen, Beijing, China) from respective tissues according to the manufacturer's instructions. RNase-free DNase I (Tiangen, Beijing, China) was used to remove genomic DNA. First-strand cDNA was synthesized using the PrimeScript II 1st Strand cDNA Synthesis Kit (Takara, Beijing, China). For a standard qRT-PCR technical application, the samples were loaded to a TB green Premix ExTap[TM] (Tli RNaseH Plus) (TaKaRa, Shiga, Osaka, Japan). The PCR reactions were run on Step One Plus[TM] Real-Time PCR System (AB, USA) with a three-step PCR procedure with the following cycling parameters: 95 °C for 30 s, followed by 95 °C for 5 s and 60 °C for 30 s, for 40 cycles, and a melt cycle from 65 °C to 95 °C. The amplification efficiency of the primers was evaluated with dilutions of cDNA, producing an $R^2$ value $\geq$ 0.99. Relative expression level of each target gene was calculated by normalization against the geometric mean [42] of transcript levels of three reference genes: *PtUBIC* (Potri.006G205700), *Ptβ-Actin* (Potri.019G006700), and *PtEF1α* (Potri.006G130900). Primers used for specific genes in this study are listed in Table S1.

### 2.6. In-Gel Activity of AspAT Enzymes

Determination of the enzyme activity was performed according to a previous study [26]. Briefly, poplar tissues were firstly ground in liquid nitrogen. The resulting powder was mixed with fresh extraction buffer (50 mM Tris-HCl pH 7.5, 10% glycerol, 0.1% Triton X-100 and 1 mM PMSF), followed with clarification by centrifugation at 13,000× *g* for 10 min (4 °C). Equal amounts (30 μg) of extracted proteins were then subjected to native polyacrylamide gel electrophoresis (PAGE) in a discontinuous system consisting of a 4% stacking gel and a 7.5% separating gel and initiated in a running buffer (25 mM Tris-HCl and 250 mM Gly, pH 8.3). Staining for AspAT/PAT activities was conducted in two steps including gel incubation for 30 min in a reaction mixture (0.1 M Tris-HCl buffer, pH 7.5, 0.2 M L-aspartic acid, 0.2 M 2-oxoglutarate, 0.1 M Ca(NO$_3$)$_2$, and 0.5% (w/v) PVP-40), followed by staining in a fast violet blue solution (2 mg mL$^{-1}$) for about 10 min. The intensity of the purple of AspAT/PAT activity bands was quantified by ImageJ software (https://imagej.nih.gov/ij/index.html).

## 3. Results

### 3.1. Genome-Wide Identification of AspATs in Populus and Other Plant Species

To identify the *AspAT* encoding genes in the genome of *P. trichocarpa*, a homology-based search using reported *AspAT* in *Arabidopsis* and rice as queries was conducted in Phytozome v12.1. A total of 10 putative isogenes were retrieved and proposed to be *AspAT* candidates after removal of the redundant sequences. With a manual reannotation, these isoforms were confirmed and designated *PtAspAT1* to *PtAspAT10* on the basis of the increasing chromosome number in accession ID (Table S2). Of note, initial BLAST with amino acid sequences resulted in findings of two genes annotated as Potri.007G08840 and Potri.T079800, showing 100% identity within the overlapped fragments. Further alignment of the genomic sequences (to upstream 2.5 kb) displayed a high identity (99%), indicating the same gene. The locus name, physical location, transcripts, CDS, protein length, MW, pI, and the deduced subcellular targeting are updated in Table S2. It was found that seven *PtAspATs* were identified as containing transcript variants. The protein length of PtAspAT varied from 304 to 481 amino acid residues, with the molecular weights (MW) ranging from 33.1 to 53.3 kDa. Most *PtAspATs* showed basic theoretical pI except for *PtAspAT8*. *In silico* prediction of their targeting organelles suggested that *PtAspATs* were localized to the chloroplasts (*PtAspAT1*, *5*, *6*, and *9*), cytosols (*PtAspAT3*, *4* and *10*), and mitochondria (*PtAspAT2* and *PtAspAT7*).

### 3.2. Evolutionary Relationships of AspATs between Populus and Other Species

To gain insight into the phylogenetic relationships, homologous *AspATs* were identified further in another 11 plants including seven dicotyledons: *A. thaliana*, *E. grandis*, *G. max*, *G. raimaondii*, *M. truncatula*, *N. benthamiana*, and *S. lycopersicum*; three monocotyledons: *O. sativa*, *S. bicolor*, and *Z. mays*, alongside one gymnosperm: *P. taeda*. Total numbers of *AspAT* homologs identified in various plant species varied from 5 to 11 (Table 1). Phylogenetic analyses revealed that 10 *PtAspATs* were clustered into two major subfamilies, namely Iα and Iβ (Figure 1). Iα subfamily is composed of eubacteria and eukaryote types of *AspATs* that can be further clustered into three groups: Iα-A, Iα-B, and Iα-C. It was found that eight members of *PtAspATs* belonged to the Iα subfamily including three members (*PtAspAT3*, *4*, and *10*) in Iα-A, three members (*PtAspAT2*, *8*, and *7*) in Iα-B, and two members (*PtAspAT5* and *PtAspAT9*) in Iα-C; however, the remaining two members (*PtAspAT1* and *PtAspAT6*) were grouped together in the Iβ subfamily containing prokaryotic AspATs, a bifunctional enzyme displaying both typical AspAT and PAT activities (Figure 1).

**Table 1.** Numbers of *AspAT* encoding genes identified in a variety of plant species.

| Plant Species | Numbers of AspATs in Subfamilies | | | | Total |
|---|---|---|---|---|---|
| | Iα-A | Iα-B | Iα-C | Iβ | |
| *Populus trichocarpa* | 3 | 3 | 2 | 2 | 10 |
| *Arabidopsis thaliana* | 3 | 1 | 1 | 1 | 6 |
| *Eucalyptus grandis* | 3 | 2 | 2 | 2 | 9 |
| *Glycine max* | 2 | 3 | 2 | 4 | 11 |
| *Grossypium raimondii* | 3 | 2 | 2 | 2 | 9 |
| *Medicago truncatula* | 1 | 2 | 1 | 7 | 11 |
| *Nicotiana benthamiana* | 2 | 1 | 1 | 1 | 6 |
| *Oryza sativa* | 1 | 2 | 1 | 2 | 6 |
| *Pinus taeda* | 1 | 1 | 1 | 2 | 5 |
| *Sorghum bicolor* | 1 | 1 | 1 | 3 | 6 |
| *Solanum lycopersicum* | 2 | 2 | 1 | 1 | 6 |
| *Zea mays* | 1 | 1 | 2 | 2 | 6 |

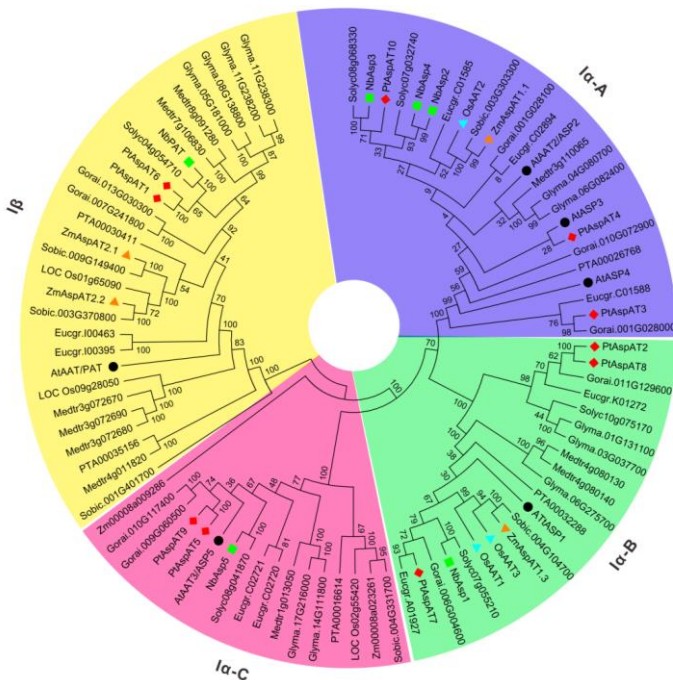

**Figure 1.** Phylogenetic relationship between *Populus* and other plant species. Full-length protein sequences of *AspATs* from 12 plant species were aligned by ClustalW. An unrooted tree was constructed by MEGA7 using the neighbor-joining method. The percentage of replicate trees in which the associated taxa clustered together in the bootstrap test (1000 replicates) is shown next to the branches. *PtAspATs* are labeled with a red solid diamond. Some AspATs reported previously in Arabidopsis, tobacco, rice, and maize were marked in various colorful shapes. Pt (*P. trichocarpa*), At (*A. thaliana*), Eucgr (*E. grandis*), Glyma (*G. max*), Gorai (*G. raimaondii*), Medtr (*M. truncatula*), Nb (*N. benthamiana*), Os (*O. sativa*), Pta (*P. taeda*), Sobic (*S. bicolor*), Solyc (*S. lycopersicum*), Zm (*Z. mays*).

### 3.3. Chromosomal Location, Genomic Structure, and Cis-Elements

Analyses of chromosomal distribution indicated that 10 *PtAspATs* were mapped on six out of the 19 chromosomes. Four *PtAspATs* were anchored on chromosome 6 and two members were located on chromosome 18, whereas others were individually distributed on chromosomes 5, 7, 14, and 16, respectively (Figure 2a). To obtain a view of the genomic patterns, the organization of exon/intron and duplication event was analyzed thereafter. As shown in Figure 2b, the number of exons varied from 10 to 14. *PtAspATs* in the Iα-A subgroup appeared to contain the largest number of exons (12–14). Interestingly, in contrast to similar exon lengths of *PtAspATs* within each cluster, the intron lengths varied significantly. *AspAT7* was identified with the maximal length of DNA sequence owing to the long sizes of its introns. Based on the phylogenetic relationships, the duplication events were proposed to occur in the *Populus* genome. Non-synonymous to synonymous substitution ratios (Ka/Ks) were determined for the paralogous *PtAspAT* gene-pairs [43], indicating that they might have evolved as a consequence of segmental duplication. The divergence time of the paralogous gene pairs of *AspAT1/6*, *AspAT4/10*, and *AspAT5/9* were estimated to be about 9.53, 1.9, and 5.99 million years ago, respectively (Table S2).

To obtain an overview of the regulatory *cis*-acting elements involved in the responsiveness of abiotic and abiotic stresses, the 1.5 kb upstream sequences from each *PtAspATs* were programed in the PlantCARE server. As shown in Figure 3, potential environmental factor-related *cis*-regulatory elements were predicted to be correlated with light, anaerobic induction, and wounding response, which were most widely spread in promoters of *PtAspATs*. Sulfur-responsive *cis*-elements (GAGAC) were identified in nine members of *PtAspATs*. Among the phytohormone regulatory elements, abscisic acid (ABA)-responsive elements (ABRE) and gibberellin (GA)-responsive elements (GARE-motif,

P-box, and TATC-box) were identified in eight *PtAspATs*. Furthermore, ethylene-responsive elements (ERE), methyl jasmonate (MeJA)-responsiveness elements (TGACG and CGTCA), salicylic acid (SA)-responsive elements (TCA), drought-responsive elements (CAACTG and ACCGAGA), and sucrose-responsive elements were also predicted to be specifically distributed in a small number of *PtAspATs*. Additionally, some transcription factor (TF) binding sites including the Dof-core site (AAAGAT) involved in C metabolism and the MYB binding site (MBSI) involved in flavonoid biosynthesis were also predicted as shown in Table S4.

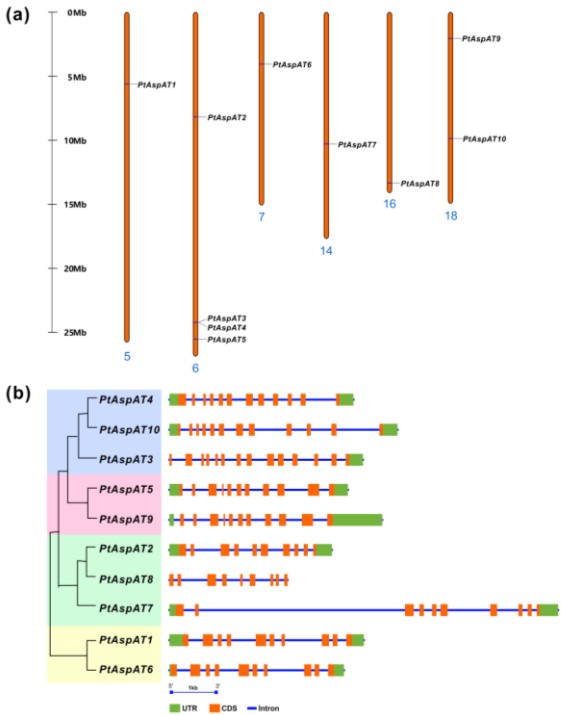

**Figure 2.** Chromosomal distribution and the genomic structures of *PtAspATs*. (**a**) Ten *PtAspATs* were anchored on six chromosomes. (**b**) Gene structures showing the exon intron organization of *PtAspATs* that were analyzed by the online tool GSDS. Lengths of the exons and introns of *PtAspATs* are displayed proportionally to the scale on the bottom. The classification of *PtAspATs* is indicated by the phylogenetic tree on the left.

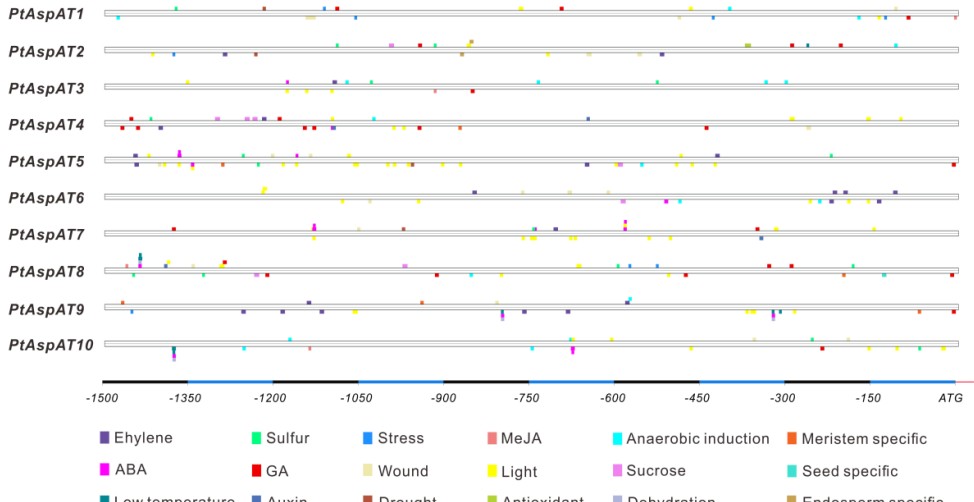

**Figure 3.** Prediction of the *cis*-regulatory elements in the promoters. Upstream 1500 bp sequences of each gene promoter were analyzed in the PlantCARE server. Stress-related *cis*-regulatory elements are spotted in different colors.

### 3.4. Conserved Motif and Domain of PtAspAT Proteins

To discover the conserved motifs of PtAspATs, the full-length protein sequences were analyzed using the MEME program. As shown in Figure 4a, a total of 15 individual motifs were characterized and the length of the amino acid residues varied from 11 to 50 (Table S5). PtAspATs in Iα contained motifs 1, 3, 5, and 6, while PtAspATs in Iβ contained motif 9, 10, 12, 13, and 15. Generally, PtAspATs in the same cluster shared similar motifs. However, an exception was found for PtAspAT8, which lacked motifs 2, 4, and 7 in comparison with other members in the Iα-B group. Furthermore, motif 14 was identified particularly for PtAspAT5 and PtAspAT9, which was clustered in the Iα-C group. Apparently, members in the Iα and Iβ subfamilies exhibited a distinct motif distribution and arrangement scheme; however, PtAspATs within the same cluster showing conserved patterns are correlated with their phylogenetic relationships and gene classification.

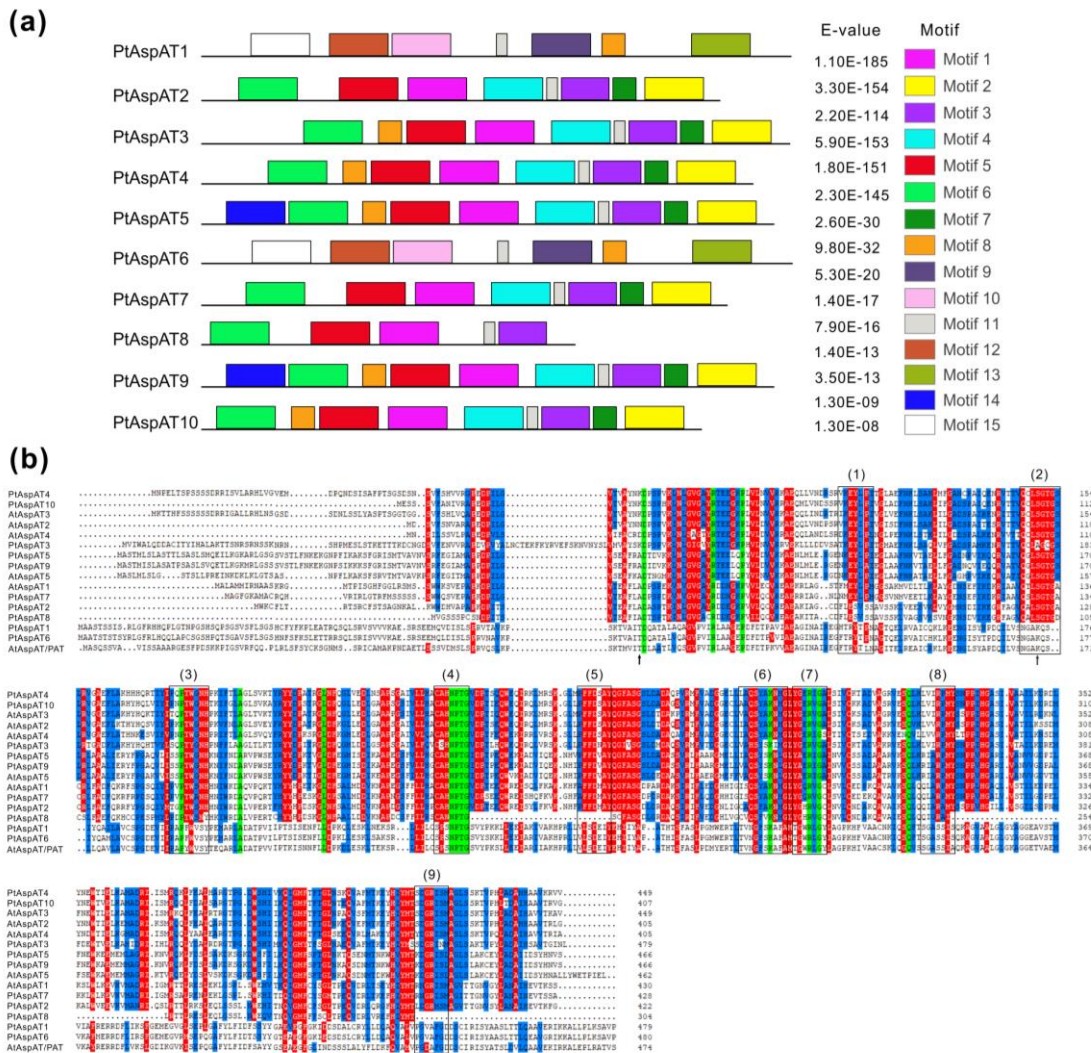

**Figure 4.** Conserved motifs and amino acid residues of AspATs. (**a**) MEME analysis showing distributions of 15 conserved motifs in 10 PtAspATs that were demonstrated with a colorful box. Detailed information is provided in Table S2. (**b**) Conserved patterns of AspATs in *Arabidopsis* (AtAspAT1 to AtAspAT5, and the bifunctional enzyme AtAspAT/PAT) and *Populus*. Nine fragments involved in binding with PLP (pyridoxal 5′-phosphate) or dicarboxylic substrate are highlighted in black boxes. The three residues reported in recognition of prephenate and Asp substrates of the Iβ subfamily are marked with black arrows.

To further evaluate the conserved patterns of AspATs, multiple sequence alignment was conducted by Clustal X2.1. Analyses of the conserved regions of AspATs between *Arabidopsis* and *Populus* resulted in the identification of nine conserved fragments (Figure 4b). These conserved fragments have been characterized widely in rice, maize, wheat, and millet [44]. Each fragment contained at least one amino acid residue that was associated with the molecular interaction with PLP (pyridoxal 5′-phosphate) or dicarboxylic substrate. Three amino acid residues (Thr84, Lys169 and Arg445, marked with arrows) were identified in PtAspAT1 and PtAspAT6, which have been demonstrated to exert essential roles in the recognition of prephenate and a specific substrate of Asp in *Arabidopsis* [45]. These conserved amino acid residues in AspATs present exclusively in the Iβ subfamily, which may be tallied well with the functional divergence between Iβ and Iα.

### 3.5. Expression Profilings of PtAspATs in Various Tissues, and in Response to N Feeding

To gain insights into the spatiotemporal expression patterns, transcripts of *PtAspATs* were initially analyzed in *P. trichocarpa* by using transcriptomic data, which were retrieved from the Phytozome (v12.1) and GEO datasets (GSE81077), including 18 tissues and five vascular cell types, respectively. As shown in Figure 5a,b, RNA-seq data generated in the heat map demonstrated a significant expression variation of *PtAspATs* in all vegetative and reproductive tissues. *PtAspAT1* and *PtAspAT10* showed transcript abundance in all vegetative tissues and the highest levels were observed both in the roots and stems. Their transcripts were substantially high in fiber and phloem, respectively (Figure S1). *PtAspAT4* was expressed specifically in the roots and stems as well as in the male flowers. *PtAspAT9* was highly expressed in the leaves and dormant buds. Both transcripts of *PtAspAT3* and *PtAspAT5* were predominantly detected in the root tips. Expression of *PtAspAT6* showed very low levels at all developmental stages, and its highest expression level was observed in leaves. *PtAspAT2* appeared to be expressed only in the stems. Additionally, transcripts of *PtAspATs* were further analyzed in the roots and stems under feeding of different N sources including ammonia (($NH_4$)$_3PO_3$), nitrate ($KNO_3$), and urea (Figure S2). Interestingly, in the roots, *PtAspAT4* was induced significantly upon the addition of all N sources and *PtAspAT3* was promoted by ammonia and nitrate. In the stems, *PtAspAT3* showed a high responsiveness to all types of N feedings, whereas *PtAspAT4* appeared to be merely affected by the addition of ammonia. Notably, transcripts of *PtAspAT8* were almost undetected in most of the selected tissues nor upon altered N sources.

To verify the tissue-specific expression of *PtAspATs*, the qRT-PCR assay was performed in four selected vegetative tissues (mature leaves, young leaves, stems, and roots). Expression of *PtAspAT8* is not presented in Figure 5c, given its extremely low signal. It was shown that plastidic *PtAspAT1* was highly expressed in the young leaves and stems. *PtAspAT10* showed the highest expression levels in the roots and mature leaves, confirming the analyses of its transcript abundance in RNA-seq (Figure 5a). Additionally, specific expressions in the roots were characterized for four genes (*PtAspAT2*, *4*, *5*, and *6*). Two genes (*PtAspAT7* and *PtAspAT9*) showing similar patterns were predominantly expressed in the leaves, which fitted well with the transcriptomic data (Figure 5a). Given that the transcript variants were identified in several genes of *PtAspAT*, the alternative splicing (AS) events were examined in *Populus*. Expression analysis was further conducted in the tissues as selected above. One of the *PtAspAT1* transcript variants, *PtAspAT1-1* appeared to be specifically expressed in the roots, whereas *PtAspAT1-2* and *PtAspAT1-3* displayed almost identical expression patterns, showing high expressions in leaves (Figure S3). Interestingly, for *PtAspAT9*, similar patterns of three transcript variants (*PtAspAT9-1*, *9-2*, and *9-4*) were observed to be highly expressed in leaves; however, the expression levels of *PtAspAT9-3* were extremely low. As no specific primers were programed for specific targeting of other genes containing multiple transcripts, examination of their expressions was not successful (data not shown).

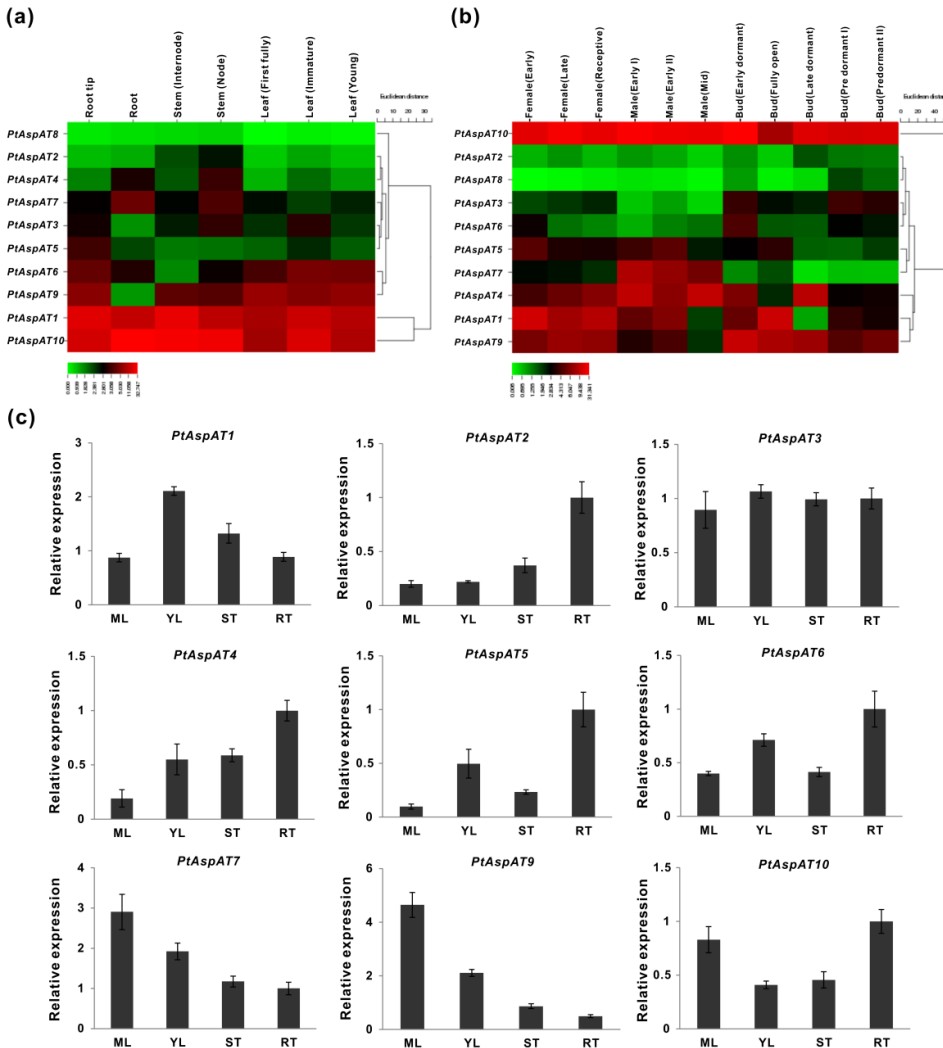

**Figure 5.** Expression profiles of *AspATs* in various tissues of *P. trichocarpa*. (**a**,**b**) Transcriptomic analyses in heat map showing the transcript abundance of *PtAspATs* in vegetative and reproductive tissues. (**c**) Tissue-specific expression of *PtAspATs* in ML (mature leaf), YL (young leaf), ST (stem), and RT (root). The RNA-seq results were given in FPKM (fragments per kilobase per million reads) expression values. Data represent mean values ± standard error (SE) of at least three independent biological replicates for qRT-PCR. *PtActin*, *PtUBIC*, and *PtEF1α* were used as reference genes.

### 3.6. Expression Evaluation of AspATs in Poplar Clone 'Nanlin895'

Poplar 'Nanlin895' is a hybrid clone and exhibits superior growth performance and manipulable features for genetic transformation when compared with other locally planted *Populus* cultivars [46]. Tissue-specific expression patterns of *AspATs* were first validated in six vascular tissues by qRT-PCR (Figure 6a). As shown in Figure 6b, a relative constant expression of *AspAT1* was detected in all tissues of the poplar '*Nanlin895*'. Along with *AspAT7*, *AspAT4* was predominantly expressed in roots and leaves. A similar pattern was identified for three genes (*AspAT2*, *3*, and *10*), whose transcripts were preferentially accumulated in the stems, and were moderately expressed in other tissues (Figure 6b). Besides, *AspAT5* and *AspAT9* were identified with the highest expression levels in mature and young leaves, followed by the roots. Additionally, it was found that *AspAT6* showed specific expression in roots, stems, and petioles. Unfortunately, no transcripts of *AspAT8* were examined in any tissues as the signal was lower than the detected level. As noted, PtAspAT8 had a distinct protein feature and its sequences were less conserved to other PtAspATs (Table S1, Figures 2b and 4), we assumed that *PtAspAT8* might evolve a distinct function. This notion was supported by the finding that *cis*-acting

regulatory element (RY-element) associated with seed-specific regulation was characterized in the *PtAspAT8* promoter region (Figure 3).

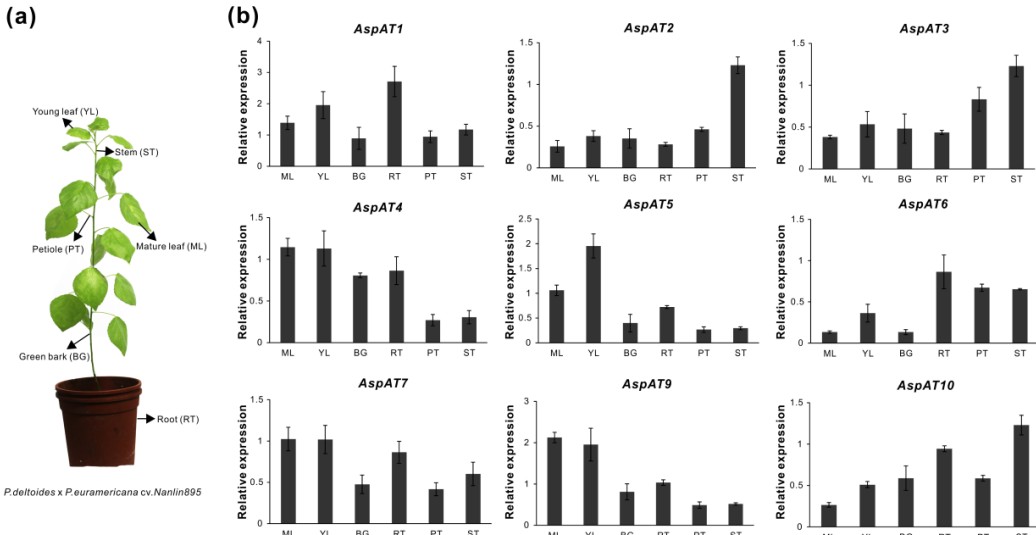

**Figure 6.** Expression analysis of *AspAT* genes in various tissues of poplar '*Nanlin895*'. (**a**) Samples were harvested from ML (mature leaves), YL (young leaves), BG (green barks), RT (roots), PT (petioles), and ST (stems). (**b**) The expression analysis was conducted by qRT-PCR. Relative expression of ML was set as 1. Data represent mean values ± SE of at least three independent biological replicates for qRT-PCR. *PtActin*, *PtUBIC*, and *PtEF1α* were used as reference genes.

Given the findings of the spatiotemporal expression patterns of *AspATs* identified in the poplar '*Nanlin895*', the impacts on the transcripts of the mature leaves and roots upon the N perturbations were subsequently investigated. Plants were cultivated with supplies of different concentrations of N nutrients ($NH_4NO_3$). Normal N (2 mM) was used as the control in parallel with N starvation (0 mM) and N excess (10 mM). Growth phenotypes were visualized during the time courses (two, three, and four weeks) of N treatments. Notably, four weeks of stressful treatments were able to sufficiently alter the morphologies in the source leaves and sink roots particularly under N starvation. As shown in Figure 7a,b, compared with the control, a total of six *AspATs* appeared to be significantly affected in either the leaves or roots, in response to N starvation. In the leaves, five genes (*AspAT1*, *2*, *5*, *6*, and *7*) were markedly induced concurrent with downregulation of *AspAT4*; however, in the roots, only *AspAT1* and *AspAT6* were drastically promoted; meanwhile, four other genes (*AspAT1*, *4*, *9*, and *10*) were identified with remarkable decreases. In contrast to the control, excessive N feeding resulted in significant suppressions of a large number of *AspATs* transcripts in the roots, whereas *AspAT10* was surprisingly induced in the leaves. Moreover, some genes characterized with high responsiveness to N deletion prompted us to further examine their transcripts in seasonal senescent leaves. Interestingly, except for *AspAT4*, most gene expression displayed drastic decreases during three selected senescent stages, suggesting that senescence-triggered N deletion directly leads to the suppression of *AspAT* expressions (Figure 7c). Nevertheless, it was hypothesized that the regulatory mechanisms underlying leaf senescence were not as common as simply removing the N nutrients from the medium.

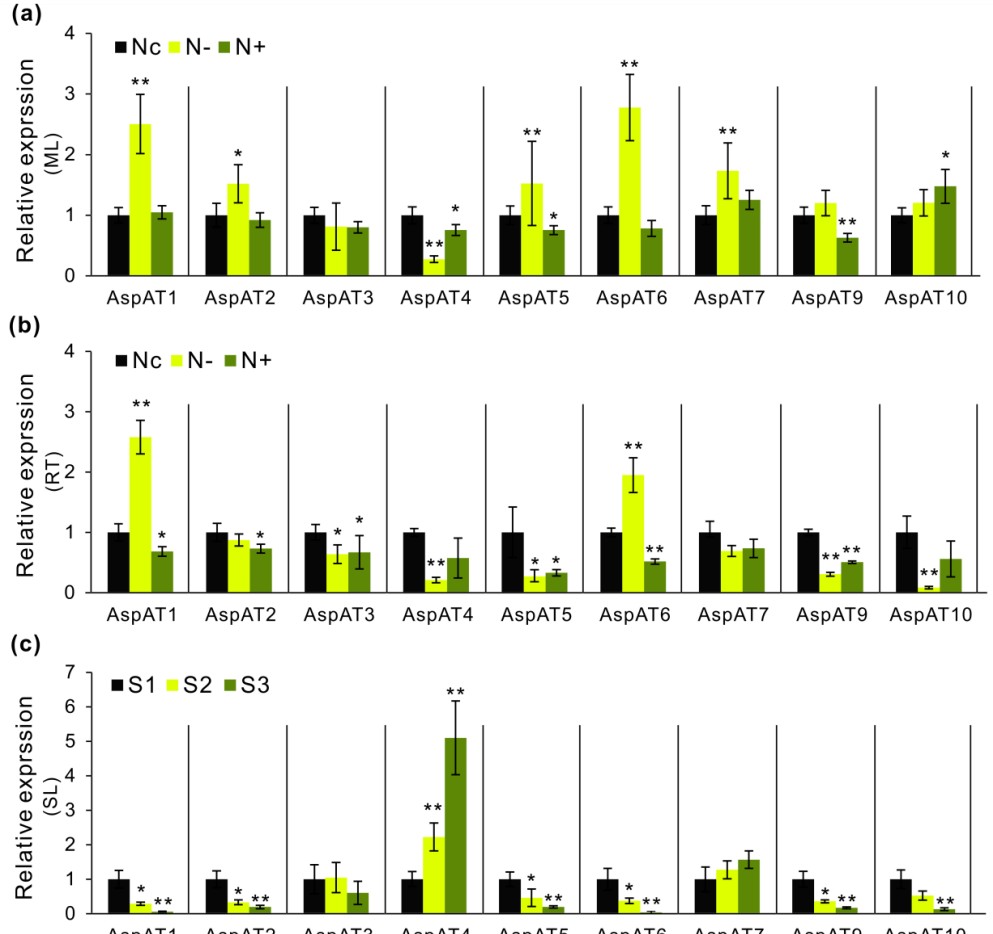

**Figure 7.** Transcript analyses of *AspATs* in response to the perturbation of N nutrients and seasonal senescence. (**a**,**b**) qRT-PCR analyses showing the effects of *AspATs* expression upon the altered N nutrient (NH$_4$NO$_3$), N– (0 mM), Nc (2 mM), and N+ (10 mM). Respective samples were harvested in poplar '*Nanlin895*' after four weeks of N treatments. (**c**) qRT-PCR analyses showing the effects of *AspATs* expression at three stages of senescent leaves (SL) in the poplar '*Nanlin895*'. Relative expression of the control (Nc or S1) was set as 1. Data represent mean values ± SE of at least three independent biological replicates for qRT-PCR. *PtActin*, *PtUBIC*, and *PtEF1α* were used as reference genes. Asterisks indicate significant differences in comparison with the control using the Student's *t*-test: *** *p* < 0.001, * *p* < 0.05.

### 3.7. Enzyme Activities in Native Gels

In an attempt to determine the effects on AspAT isoenzymes activities in vivo upon the aforementioned perturbations of N nutrients, extracted protein fractions were electrophoresed on native gels for the measurement of enzyme activities. Three major bands visualized from the native gel indicated three AspAT components (Figure 8a). To analyze the in-gel enzyme activities, the intensities of the stained band signal were quantified by ImageJ. As shown in Figure 8a, a large proportion of plastidic AspAT (Cp) activities were detected preferentially in mature and young leaves, and concurrently, the mitochondrial AspAT (Mt) activities were identified predominantly in the roots. Cytosolic AspAT (Cy) displayed a similar distribution tendency to plastidic AspAT (Cp), but its activities were much weaker in barks, petioles, and stems, suggesting that cytosolic isoenzyme was a minor component of AspATs (Figure 8b). In contrast to the control, in mature leaves, it was found that N nutrient starvation caused drastic decreases of enzyme activities of all three AspAT components, whereas these enzyme activities were almost unaffected in response to excessive N nutrients, suggesting a higher capacity of N utilization and recycling for plant growth (Figure 8c and Figures S4 and S5). Interestingly,

mitochondrial AspAT activities were significantly suppressed in the roots when N was removed from the medium; however, Cp and Cy AspAT activities were not significantly altered regardless of the over addition or removal of N in the medium (Figure 8d). Furthermore, consecutive decreases of activities for all AspAT components were observed in three selected stages of leaf senescence, indicating a correlation between enzyme activities and leaf aging (Figure 8e).

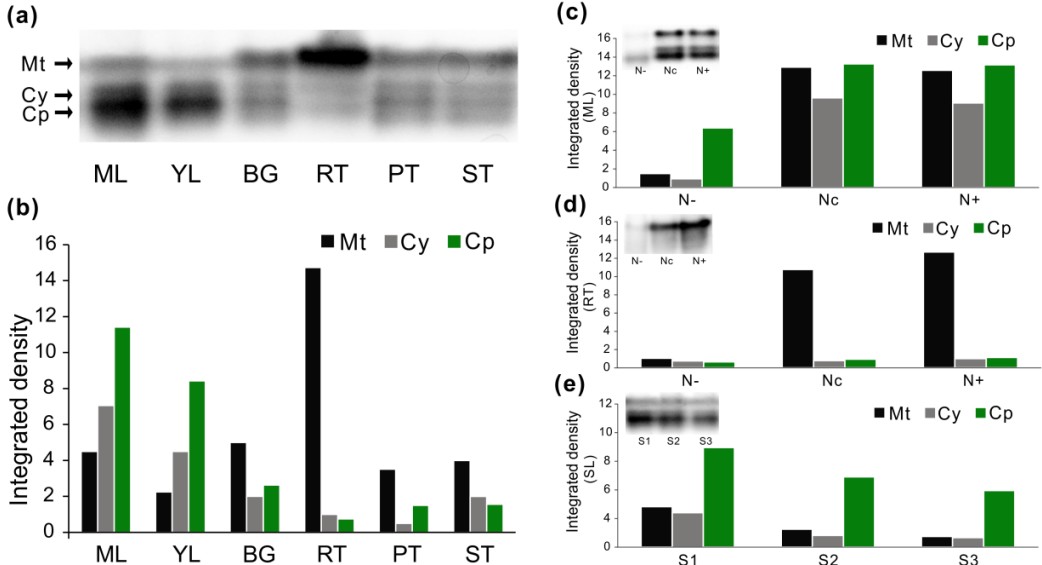

**Figure 8.** Effects of AspAT activities in response to the perturbation of N nutrients and seasonal senescence. (**a**,**b**) Determination of AspAT activities in various tissues of poplar '*Nanlin895*' in native gel and signal strength was quantified by ImageJ. ML (mature leaves), YL (young leaves), BG (green barks), RT (roots), PT (petioles), and ST (stems). (**c**,**d**) Effects of enzyme activities upon perturbations of N nutrients in roots and leaves, respectively. (**e**) Effects of enzyme activities in leaves upon seasonal senescence. Samples were harvested as described in Figure 6. Each well was loaded with an equal amount (30 µg) of extracted protein and enzyme activities were assayed in native PAGE gels. The enzyme activity data represent one of three independent biological replicate s and every replicate showed similar patterns.

## 4. Discussion

### 4.1. AspAT *Gene Family in* Populus

*Populus* is a model species of perennial woody plants. Even if the entire genome sequence of *Populus* has been released for a decade, a thorough survey of *PtAspATs* has not been reported hitherto. One of our primary objectives was to gain novel insights into the molecular aspects of the *AspATs* gene family in *Populus*. An initial comprehensive genome-wide analysis demonstrated that a total of 10 *PtAspATs* were identified from the recent released *Populus* genome (v3.0) in Phytozome 12.1. Numbers of *AspAT* homologs were identified in 12 plant species, suggesting that the *AspATs* were encoded by a small gene family. Along with other plants species, phylogenetic analyses showed that *PtAspATs* consisted of two major subfamilies, namely Iα (A, B, and C) and Iβ (Figure 1). Members of the Iα subfamily displayed identities varying from 44% to 73%; however, Iα and Iβ only shared identities of 21%–28%, which was in line with the previous report [2]. Nevertheless, *PtAspAT1* and *PtAspAT6* showed high identities (>90%) to each other and also among various plants within the Iβ subfamily, reflecting a more conserved pattern of *Iβ AspAT* subfamily.

*PtAspATs* in the Iα subfamily were further clustered into three subgroups: Iα-A, Iα-B, and Iα-C. Interestingly, they were postulated to be tightly correlated with the predicted subcellular targeting, since *PtAspATs* in Iα-A were deduced in cytosols, members in Iα-B were mostly localized to plastids, and Iα-C contained mitochondrial *PtAspATs* (Figure 1 and Table S2). In addition, gene structure

analyses suggested that *PtAspATs* showed similar patterns of exon/intron organization within the same subgroup (Figure 2a). The genome duplication event was considered to facilitate organisms for adaption to environmental changes during millions of years of evolution, especially for woody perennials [31]. Estimation of the duplication event by the Ka/Ks ratios suggested that negative or purifying selection pressure had a major contribution to the maintenance of the function of *PtAspATs* in the *Populus* genome (Table S3). Three gene pairs (*AspAT1/6*, *AspAT4/10*, and *AspAT5/9*) may undergo multiple rounds of segmental duplication, prompting their special physiological roles associated with complicated mechanisms of adaptation to stressful stimuli. The existence of diverse groups of *cis*-regulatory elements is indicative of the very dynamic regulation of *PtAspATs*. Further identification of prevalent diverse *cis*-regulatory elements related to phytohormones (ABA, GA, MeJA, and Auxin), abiotic stress (wounding, low temperature, antioxidant, and drought), and nutrients, along with TF binding sites associated with stress response, gave us a strong hint that molecular regulation of *PtAspATs* relies significantly on the crosstalk between phytohormones, stress, and nutrient compounds (Figure 3 and Table S4). Furthermore, the conserved patterns of the AspAT protein sequences were analyzed via the multiple sequence alignment between *Populus* and Arabidopsis, resulting in the characterization of 15 conserved motifs and nine typical features for AspATs. Three key amino acid residues (Thr84, Lys169 and Arg445) were postulated as determinant in distinguishing Iα from Iβ (Figure 4b). More recently, the specificities of three amino acid residues in recognition of keto and amino acid substrate have been verified by analyzing the protein crystal structure and site-directed mutation, suggesting that plant derived-AspAT/PAT encoding genes most likely evolved from prokaryotic ancestors [45,47].

### 4.2. Mining Expression Profiles of AspATs in Tissues of Different Poplar Clone

Recent advances in high-throughput sequencing have boosted a significant innovation in functional genomic research. To evaluate whether a gene may be a candidate involved in a specific metabolic process, one strategy is to investigate the patterns of gene expression [48]. Thus, to understand how *AspATs* respond to N, transcription profiling technology was used to analyze gene expression. The spatiotemporal expressions of *AspATs* were initially examined by RNA-seq data, which were further validated by qRT-PCR assay (Figure 5). The depicted expression patterns of individual *PtAspATs* were postulated to match their proposed functions in particular developmental stages, which is in accordance with the expression patterns of homologs in other plant species [1,13–15]. Interestingly, the evaluation of qRT-PCR in *P. trichocarpa* revealed that a few genes showed similar expression patterns in all selected tissues and/or certain types of tissues and cells, such as *PtAspAT2* and *PtAspAT4*, *PtAspAT5* and *PtAspAT6*, and *PtAspAT7* and *PtAspAT9*, reflecting that their co-expressions may be involved in functional complementation during plant growth and development. This result is in agreement with a recent report, showing co-expression networks of structural genes in wood formation [40]. The expressions of plastidic *PtAspATs* are much more complicated, since two types (Iα-C and Iβ) of *AspAT* isozymes co-exist in plants [14]. Surprisingly, tissue-specific expression of gene pairs with segmental duplication also showed differential expression patterns (Table S2) and therefore, we hypothesized that this phenomenon was caused by the overlapped expression levels of transcript variants. In non-woody plants, the AS of genes is considered as an efficient way for generating variation in protein structure, functional diversity, and stress acclimation, and has been observed among various tissues, cell types, and different treatments [49,50]. In our study, the experimental evaluation of transcript variants of *PtAspAT1* and *PtAspAT9* corroborated that AS was prevalent and a reason for the differential expression within the duplicated *AspATs* in *Populus*, pointing to transcriptional regulation as the primary mechanism underlying the modulation of AspAT function in N metabolism.

Verification of *PtAspAT* expressions in major vegetative tissues of *P. trichocarpa* prompted us to identify *AspAT* isogenes and their expression variations in the poplar clone 'Nanlin895', a widely planted tree in south China owing to the fast growth and easy manipulation of transformation [46,51]. Analyses of tissue-specific expression in the 'Nanlin895' revealed that nine *AspAT* isogenes showed

transcript abundance in various vascular tissues, whereas these genes expression profiles appeared dissimilar to our previous presentation of transcript abundance of nine *PtAspATs* in respective tissues of *P. trichocarpa*, reflecting the genetic variation between two different poplar clones [52]. Nevertheless, similar phenomena were observed in the poplar clone 'Nanlin895' that showed an alike transcript tendency for three pairs of genes (*AspAT2* and *3, AspAT4* and *7*, and *AspAT5* and *9*), confirming again that the postulated co-expression of *AspATs* may contribute to the functional complementation in normal plant growth (Figure 6). Although gene duplication events occurred commonly in *Populus*, expression in the same tissue or at a particular developmental stage were not completely identical for the three *PtAspAT* gene pairs with segmental duplication except for plastidic *AspAT5* and *AspAT9* (Figures 2b and 5); again our results suggest that AS may facilitate the overlapped expression levels of the transcript variants in a certain tissue particularly under stressful conditions. Overall, expression analyses of *AspAT* isogenes in specific types of tissue are beneficial for the subsequent exploration of transcriptional and post-transcriptional interference in response to stress cues.

### 4.3. Modulation of AspAT *Transcripts and Activities upon N Alteration*

N is a major macro nutrient for plant growth and development as it can be integrated into multiple metabolic processes and drives the biosynthesis of a large number of molecules, such as amino acids, proteins, pigments, and phytohormones as well as secondary metabolites [53]. Uptake and utility of N are required to improve the biomass and productivity of crops including fast-growing woody plants like *Populus* [22]. To date, a significant remaining challenge for the tree functional study of a particular gene family is to surpass the genomic overview of candidate lists, and link transcript abundance and genetic variation with the enzyme kinetics [54]. As one of the initial steps, we investigated the influence of *AspATs* transcripts in the source leaves and sink roots of 'Nanlin895' upon the perturbation of N nutrients availability and seasonal senescence (Figure 7). It was found that a much lower number of *AspAT* transcripts were significantly suppressed in the mature leaves than in the roots under excessive N feeding, indicating that such N concentration was a stressful dosage for roots but not for aerial leaf tissues. It is worthwhile noting that when we focused on the transcript responsiveness of plastidic *AspATs* in the roots upon the altered N, *AspAT1* and *AspAT6* showed marked increases of transcripts in response to N starvation, whereas by contrast, decreased transcripts upon excessive N were identified concomitantly, suggesting their functions in the maintenance of N homeostasis in sink tissues. Interestingly, two pairs of genes (*AspAT4* and *10, AspAT5* and *9*) were characterized to be drastically suppressed by both N starvation and N excess, confirming that N stress was able to trigger the co-expression of gene pairs with duplication [22]. Accordingly, seasonal senescence mediated-N starvation caused drastic suppressions of large numbers of *AspATs* gene transcripts during three senescent stages (Figure 7c). The characterization of differential expression patterns suggested different regulatory mechanisms deploying N remobilization and degradation between biological aging and artificial N deficiency. Overall, the observed fluctuation of gene transcripts pointed out that consequent up- and down-regulation of *AspAT* in the N metabolic process may lead to a significant modification of enzyme activities in different components upon altered N nutrients.

In the present study, the enzyme activities of different AspATs types were distinguished in various tissues of 'Nanlin895' by a native gel assay. As noted in the zymographic figure, the enzyme activities of plastidic and mitochondrial AspATs were relatively abundant in the leaves and roots, respectively (Figure 8a). Further in-gel analyses upon stressed N in mature leaves revealed that only N starvation could cause a significant decrease of three types of AspAT activities (Figure 8c and Figure S4). We assumed that the suppression of AspAT activities may be ascribed to the degradation of functional AspAT since the sequentially abated AspAT activities were concurrently detected in seasonal senescent leaves. Moreover, a comparison of the spatial dynamics of *AspAT* expression in parallel with enzyme activities in response to altered N hinted that the modulation of cytosolic and mitochondrial AspAT might largely depend on post-transcriptional and/or post-translational mechanisms, especially under stressful conditions. Recent reports on N-starvation in microalgae have

shown that AspATs are involved in post-transcriptional regulation of amino acid biosynthesis [55]. In maize, systematic analysis of leaf developmental gradient research also indicated that regulation of AspAT was subject to post-transcriptional control [17]. Since AspAT-derived metabolite biosynthesis is of significant importance for plant growth, the tight control of both gene expression and enzyme activities by a potential multiple layer of regulatory mechanisms appears to be necessary in response to altered N levels [56]. However, whether post-transcriptional or post-translational control of AspAT as a regulatory event in plant stress tolerance remains to be unraveled. Moreover, plenty of evidence revealed that transcript levels commonly change to a much greater extent than that of the protein levels, and with different time courses. Therefore, enzyme activities, rather than gene transcripts, are considered to be a more direct and significant indicator of the plant performance as well as responsiveness under particularly stress conditions [57–59].

## 5. Conclusions

AspAT is indeed an important aminotransferase enzyme and exerts specific functions other than merely catalyzing the formation of Asp, which is regarded as an essential intermediate for protein synthesis [60]. In spite of many advances in biochemistry achieved during the past two decades, the molecular and physiological roles of *AspAT* family genes have not been well deciphered in woody plants. The presence of AspAT isozymes at different subcellular compartments and the broad spatiotemporal expressions facilitate the functional specificity and diversity in plant growth and development as well as in response to stress stimuli and environmental cues. As yet, none of *AspAT* isogenes have been hitherto comprehensively analyzed in poplars. Here, we provided molecular and physiological information through genome-wide analyses of *AspAT* encoding genes in *P. trichocarpa*. Taken together, the novel findings of (co-)expression and enzymatic patterns of AspATs in the poplar clone '*Nanlin895*' under a N stress regime lay the theoretical foundation in unveiling the metabolic and physiological significance of *AspAT* in plant growth and stress response. Future work will aim to track the metabolite biosynthesis including Asp and its relevant molecules, and crosstalk with phytohormones to understand the roles of *AspATs* in the metabolic regulation of N and C. Accordingly, significantly induced/suppressed genes will be targeted for the in vivo functions in plant growth and upon various stressors through phenotypic analyses of overexpressing and Crispr/Cas9 mutants, which has been initiated.

**Supplementary Materials:** The following are available online at http://www.mdpi.com/1999-4907/10/5/449/s1: Figure S1: Transcriptomic analysis of *PtAspATs* in vascular tissues; Figure S2: Transcriptomic analysis of *PtAspATs* in response to various nitrogen sources; Figure S3: Expression analysis of the transcript variants of *PtAspAT1* and *PtAspAT9* in different tissues; Figure S4: Effects on enzyme activities in various tissues, N treatment and leaf senescence; Figure S5: The entire native PAGE gel images showing AspAT activities that are compatible with Figure 8. Table S1: List of primers used for qRT-PCR analysis; Table S2: List of *AspAT* candidate genes in *Populus*; Table S3: Ka/Ks analysis of the *PtAspAT* gene pairs duplication; Table S4: Transcription factors predicted in binding promotors of *PtAspATs*; and Table S5: Conserved motifs of AspATs deduced by MEME analysis.

**Author Contributions:** T.S. and M.H. designed the experiment, collected and analyzed all of the data. T.S. and M.H. prepared the initial draft of the manuscript. M.H. developed the concept and was responsible for approving the final draft of the manuscript. J.M. performed the qPCR analysis and enzyme activity assay. D.C., G.Z. and N.L. assisted J.M. with the experiment conduction. H.Z. was responsible for plant culture in vitro. M.L. did much work on the bioinformatics analysis including gene identification, conserved domain and promoter analyses, RNA-seq data collection, and heat map construction. All authors reviewed the manuscript.

**Funding:** This research was supported by the National Natural Science Foundation of China (NSFC) (31700525; 31870589), the Natural Science Foundation of Jiangsu Province (NSFJ) (BK20170921), the Scientific Research Foundation for High-Level Talents of Nanjing Forestry University (SRFNFU) (GXL2017011; GXL2017012), the Priority Academic Program Development of Jiangsu Higher Education Institutions (PAPD), and NFU Undergraduate Innovation and Entrepreneurship Training Programs (2018NFUSPITP068).

**Acknowledgments:** The authors would like to thank NSFC, NSFJ, and SRFNFU for funding this research. Our thanks go to the Key Laboratory of State Forestry Administration on Subtropical Forest Biodiversity Conservation, the Co-Innovation Center for Sustainable Forestry in Southern China, and PAPD for the instrument support.

**Conflicts of Interest:** The authors declare no conflict of interest.

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
