# Peer review of "Genome-Wide Characterization of AspATs in Populus: Gene Expression Variation and Enzyme Activities in Response to Nitrogen Perturbations"

_forests, doi:10.3390/f10050449_

Round 1

Reviewer 1 Report

This article describes studies of a small family of genes, aspartate aminotransferases, in poplar. Sequence comparisons, conserved functional domain searches, and gene expression analyses are included. 

Extensive English language editing is required throughout the article. The methods section needs much more detail. For example, line 155 mentions nitrogen treatments, but they are not described at all. Section 2.2 describes sequence similarity searches using AspATs from various plant species, but the specific numbers of the sequences obtained per species is not mentioned. When the candidate AspATs were "verified" using Pfam and NCBI, how exactly was this done?

Finally, PtAspAT8 did not appear to be expressed in any of the tested tissues, but this is not discussed. Does the apparent lack of expression indicate that this is a pseudogene?

Author Response

Reviewer 1

1. This article describes studies of a small family of genes, aspartate aminotransferases, in poplar. Sequence comparisons, conserved functional domain searches, and gene expression analyses are included. Extensive English language editing is required throughout the article. The methods section needs much more detail. For example, line 155 mentions nitrogen treatments, but they are not described at all.

R1: Many thanks for the reviewer’s comments. This manuscript has undergone English language editing by the experienced editors from MDPI as well as a native English speaking researcher (Dr. Perrin H. Betty) from University of Alberta. Nitrogen treatments in this paper were conducted according to a previous report (Luo et al., 2015) (Line157-161), please check the reference [22].

2. Section 2.2 describes sequence similarity searches using AspATs from various plant species, but the specific numbers of the sequences obtained per species is not mentioned.

R2: This comment might be interesting. Specific numbers of AspATs identified in various plant species were presented in Table 1 (Line 261).

3. When the candidate AspATs were "verified" using Pfam and NCBI, how exactly was this done?

R3: Respective genomic and protein sequence of PtAspAT candidates were verified with the Pfam (http://pfam.xfam.org/) by HMMER program (3.1b2) and NCBI (https://www.ncbi.nlm.nih.gov/) to ascertain the presence of Hidden Markov Model (HMM) profiles (PF00155.20) that correspond to Aminotran_1_2. Please check the revised sentences in Materials and Methods (Line172-175).

4. Finally, PtAspAT8 did not appear to be expressed in any of the tested tissues, but this is not discussed. Does the apparent lack of expression indicate that this is a pseudogene?

R4: Good point. Even though PtAspAT8 shows distinct features with less sequence length and conserved motifs (Figures 2b, 4, and table S1), we are not able to rule out that it might play specific roles in a particular tissue. The upcoming work focusing on the inspection of its subcellular localization and functional analyses by transgenic assay will provide more detailed information. Sentences have been inserted into the text of Results (Line418-422).

Reviewer 2 Report

Su et al. reported identification of ten AspAT paralogs and studied in-depth the expression patterns of the genes in different tissues and time, and also in a poplar hybrid clone. In addition, the authors also Overall, the story is very interesting, I recommend the manuscript to be published in Forests after answering my following questions/comments:

1. In the introduction, author need to introduce the background of “Nanlin895”, and explain the reason that clone was selected for analysis.

2. Identification of the ten AspAT.

First, I get confused about the terms isogenes (L239), isoforms (L240) etc. Do the authors mean that the ten AspAT proteins are isozymes? Or talking about paralogs?

Isoforms usually mean products from a same gene but with alternative splicing. Please clarify it.

L164 Regarding identification of the AspAT homologs, did the author use Blastp, correct? I am not sure if the genome annotation of P. trichocarpa is good. If not, the author may need to think about using tBlastN. How were the gene models corrected? Based on RNA-seq data?

3.       FPKM and qPCR

Have the author normalized the RNA-seq reads/qPCR values to any internal controls when drawing the heat maps/bar charts? If yes, how did the three reference genes were used for statistical analysis? No description in the method.

4.       Figure 4

Panel a, Can the author re-group the PtAspAT paralogs based on the motif similarities rather than the name order? For instance, put those paralogs in the same family, like Iα together, Iβ together etc.

Panel b, it is difficult to read the alignment, because of the low resolution. I suggest the author trim the unaligned amino acids at the N-terminus, and use larger fonts. 

Add bars/lines, as well as the corresponding family names on the left of the alignment figure to make the classification more clear. In the legend, the authors need to explain what does AtAspat3 AtAspAT/PAT stand for.

Figure 8. In the legend, also add the full names of ML, YL, etc.

The entire PAGE gel may need to be present in the supplemental materials.

Other minor issues,

L67, it will be more clear if the authors explain the terms Iα and Iβ in the Introduction part.

Throughout the manuscript, there are many places where poplar is misspelled. Shouldn’t be polar, correct? L386, L391, L392 etc.  

All Latin names should be in italics, including those in the references.

Author Response

Reviewer 2

Su et al. reported identification of ten AspAT paralogs and studied in-depth the expression patterns of the genes in different tissues and time, and also in a poplar hybrid clone. In addition, the authors also Overall, the story is very interesting, I recommend the manuscript to be published in Forests after answering my following questions/comments:

1. In the introduction, author need to introduce the background of “Nanlin895”, and explain the reason that clone was selected for analysis.

R1: As we have described in Results (Line408-409) and Discussion (Line553-554), Nanlin895’ is a poplar hybrid that were generated by researchers in Nanjing Forestry University and also they are widely planted in south China. In addition, in contrast to other poplar cultivars, Nanlin895’ behaved superiority with fast growth in vitro and soil culture and easy manipulation of transformation. Please find the detailed information from references list in [46] and [51].  

2. Identification of the ten AspAT. First, I get confused about the terms isogenes (L239), isoforms (L240) etc. Do the authors mean that the ten AspAT proteins are isozymes? Or talking about paralogs? Isoforms usually mean products from a same gene but with alternative splicing. Please clarify it. L164 Regarding identification of the AspAT homologs, did the author use Blastp, correct? I am not sure if the genome annotation of P. trichocarpa is good. If not, the author may need to think about using tBlastN. How were the gene models corrected? Based on RNA-seq data?

R2: Thanks for the comments. These typos have been corrected. As a matter of fact, we used both BlastP and tBlastN in the identification of AspATs as well as the processing of the gene models.

3. FPKM and qPCR

Have the author normalized the RNA-seq reads/qPCR values to any internal controls when drawing the heat maps/bar charts? If yes, how did the three reference genes were used for statistical analysis? No description in the method.

R3: Multiple reference genes used in qRT-PCR analyses and statistical calculation provides more advantages than a single reference gene. We used a popular algorithm, geNorm (https://genormNaNgg.be/) to determine the most stable reference genes from a set of tested candidate reference genes in a given sample panel. A gene expression normalization factor can be calculated for each sample based on the geometric mean of a user-defined number of reference genes (Line226-228). Please find the detailed description in the reference [42].

4. Figure 4 Panel a, Can the author re-group the PtAspAT paralogs based on the motif similarities rather than the name order? For instance, put those paralogs in the same family, like Iα together, Iβ together etc. Panel b, it is difficult to read the alignment, because of the low resolution. I suggest the author trim the unaligned amino acids at the N-terminus, and use larger fonts.

R4-1: The image quality of Figure 4 has been improved significantly by elevation of the resolution (600dpi) to keep consistence with Figure 3. Please check the uploaded manuscript.

Add bars/lines, as well as the corresponding family names on the left of the alignment figure to make the classification more clear. In the legend, the authors need to explain what does AtAspat3 AtAspAT/PAT stand for.

R4-2: This comment might be interesting. As it was described in the Introduction (Line111-116), a total of six AspAT encoding genes were identified in the Arabidopsis genome including AspAT/PAT, standing for a bifunctional enzyme with both classic AspAT activity and PAT activity (Line 68-72, Table 1, and Figure 1). The legend in Figure 4 has been updated (Line335).

Figure 8. In the legend, also add the full names of ML, YL, etc.

R4-3: The full names of ML, YL, BG, RT, PT, and ST have been inserted into the legend of Figure 8 (Line457-458).

5. The entire PAGE gel may need to be present in the supplemental materials.

R5: The entire PAGE gel has been added to the supplemental materials. Please check Figure S5.

6. Other minor issues, L67, it will be more clear if the authors explain the terms Iα and Iβ in the Introduction part.

R6: Brief explanation of Iα and Iβ was added to the text (Line 68-72).

7. Throughout the manuscript, there are many places where poplar is misspelled. Shouldn’t be polar, correct? L386, L391, L392 etc. All Latin names should be in italics, including those in the references.

R7: Many thanks for the comments and these typos have been corrected and normalized.

Round 2

Reviewer 1 Report

The revisions address my previous comments. The manuscript is much improved.

Author Response

1. The revisions address my previous comments. The manuscript is much improved.

R1: Thank you for your comments! Absolutely, the manuscript was revised significantly on the basis or your comments.